# Photo-triggered solvent-free metamorphosis of polymeric materials

Satoshi Honda [1] & Taro Toyota [1]

Liquefaction and solidification of materials are the most fundamental changes observed during thermal phase transitions, yet the design of organic and polymeric soft materials showing isothermal reversible liquid–nonliquid conversion remains challenging. Here, we demonstrate that solvent-free repeatable molecular architectural transformation between liquid-star and nonliquid-network polymers that relies on cleavage and reformation of a covalent bond in hexaarylbiimidazole. Liquid four-armed star-shaped poly(n-butyl acrylate) and poly(dimethyl siloxane) with 2,4,5-triphenylimidazole end groups were first synthesized. Subsequent oxidation of the 2,4,5-triphenylimidazoles into 2,4,5-triphenylimidazoryl radicals and their coupling with these liquid star polymers to form hexaarylbiimidazoles afforded the corresponding nonliquid network polymers. The resulting nonliquid network polymers liquefied upon UV irradiation and produced liquid star-shaped polymers with 2,4,5-triphe-nylimidazoryl radical end groups that reverted to nonliquid network polymers again by recoupling of the generated 2,4,5-triphenylimidazoryl radicals immediately after terminating UV irradiation.

[1] Department of Basic Science, Graduate School of Arts and Sciences, The University of Tokyo, 3-8-1 Komaba, Meguro-ku, Tokyo 153-8902, Japan. Correspondence and requests for materials should be addressed to S.H. (email: c-honda@mail.ecc.u-tokyo.ac.jp) or to T.T. (cttoyota@mail.ecc.u-tokyo.ac.jp)

Living systems utilize masterfully designed molecular systems to maintain biological activities. For example, sea cucumbers are interesting because some species can switch their body into a liquiform state when removed from seawater but revert to their original state following re-immersion in seawater[1]. One possible mechanism underlying this biofunction is the cross-linking and de-crosslinking of collagen microfibrils in the extra-cellular matrix by the peptides tensilin and softenin[2]. The body of a sea cucumber is hard when collagen microfibrils are crosslinked with tensilin but becomes soft upon interaction with softenin and subsequent de-crosslinking. Sea cucumber-inspired materials with tuneable stiffness have been reported[3], however, isothermal reversible liquid–nonliquid conversion (IRLNC) with artificial soft materials remains challenging.

Thermally reversible solid–liquid phase transition typically relies on molecular ordering and disordering is a common process for most organic molecules. The control of molecular ordering by photoisomerization of azobenzene-containing molecules has been described, leading to isothermal crystal melting and recrystallization in response to UV[4–13]. Although such commendable studies achieved solid–liquid transition with azobenzene-containing crystalline and liquid crystalline small molecules, these molecules do not exhibit tuneable elasticity. In contrast, the diverse range of polymers synthesized to date exhibit a wide range of tailored properties. However, attaining solid–liquid phase transition using the aforementioned mechanism seems difficult even with highly crystalline polymers because polymers are inherently a mixture of crystalline and amorphous structures. Alternatively, liquid–nonliquid conversion relying on changes in the glass transition temperature ($T_g$) of side-chain azobenzene-containing polymers has been reported[14, 15]. When *trans* to *cis* photoisomerization was conducted by irradiating UV to a $CH_2Cl_2$ solution of the *trans* azobenzene-containing polymers having $T_g$s above room temperature, the obtained *cis* azopolymers after the removal of $CH_2Cl_2$ showed $T_g$ below room temperature, attaining liquid and nonliquid states, respectively, with the same polymers. Although such formidable systems have open the door for a repeatable photochemical molding technology, development of a smart polymer system to attain IRLNC with an alternative facile and versatile mechanism remains unexplored.

Progress in polymer chemistry allows us to synthesize various nonlinear polymers. Nonlinear polymers are expected to facilitate the realization of a methodology for tuning properties of polymers without changing their chemical composition or molecular weight[16]. Among a wide variety of nonlinear polymers, one basic network architecture comprises star-like components. Various network organic and polymeric materials are produced by reactions between star-shaped molecules, for example, the synthesis of silica by a reaction between tetraethylorthosilicates[17]. In most cases, the star-shaped starting molecules in the liquid state are converted to the corresponding solid network material, but such network formation reactions are in general irreversible. To address this, we focused on a repeatable molecular architectural transformation (MAT) strategy[18–23]. Various repeatable MATs, for example, linear–star[18], linear–cyclic[19–22], three-armed star–network[23] architectures have appeared and, moreover, MAT of amphiphilic block copolymers into star and comb polymers has recently been achieved by Diels–Alder reaction[24]. Of various candidates exhibiting repeatable cleavage and reformation of covalent bonds[25] utilized as self-healing materials[26], we focused on hexaarylbiimidazole (HABI)[27] as the linking point. HABI can be synthesized by the oxidation of 2,4,5-triphenylimidazoles (lophines) to generate 2,4,5-triphenylimidazoryl radicals (TPIRs) and the subsequent coupling reaction between the generated TPIRs. The TPIR is stable even under the existence of oxygen but reacts with other TPIR to generate HABI[28]. The covalent bond between the two imidazoles in HABI is cleavable by UV irradiation, leading to the production of a pair of TPIRs, which can recouple with each other.

Here we demonstrate a solvent-free IRLNC of poly(*n*-butyl acrylate) (PBA) and poly(dimethyl siloxane) (PDMS) by photo-triggered repeatable MAT between star and network architectures utilizing the HABI chemistry.

## Results

**Photo-triggered solvent-free IRLNC by repeatable MAT of polymer materials**. We envisioned that nonliquid network and

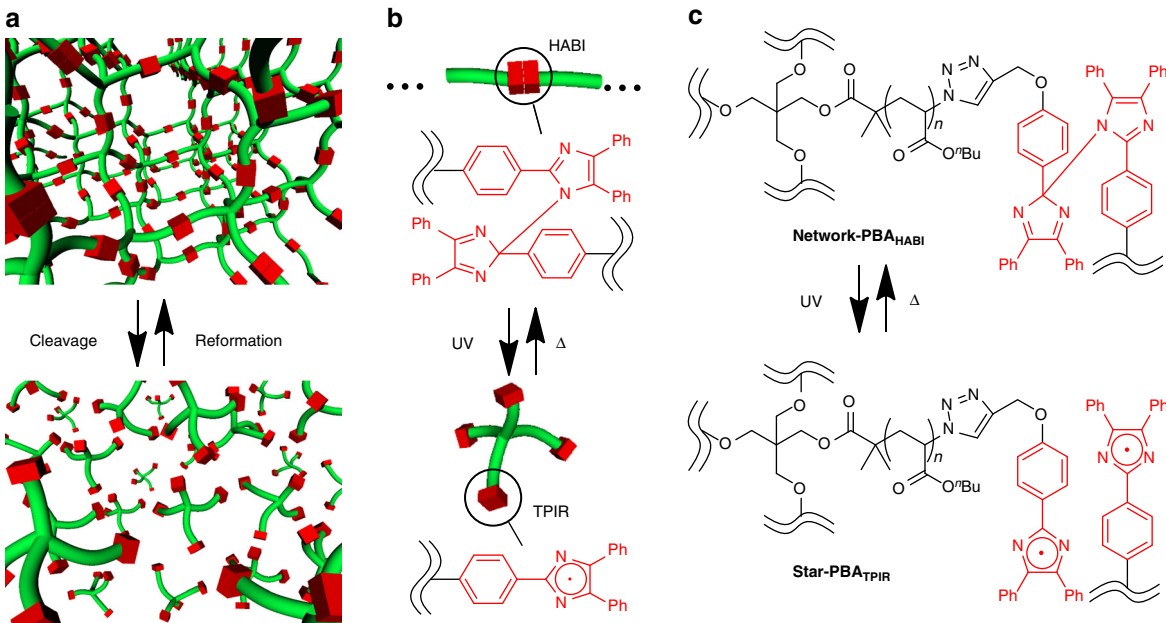

**Fig. 1** Design of star–network repeatable MAT system with PBA. Schematic representation of **a** star–network repeatable MAT, **b** cleavage and reformation of the covalent bond in HABI, and **c** repeatable MAT between **network-PBA_HABI** and **star-PBA_TPIR** based on UV-triggered bond cleavage and recoupling

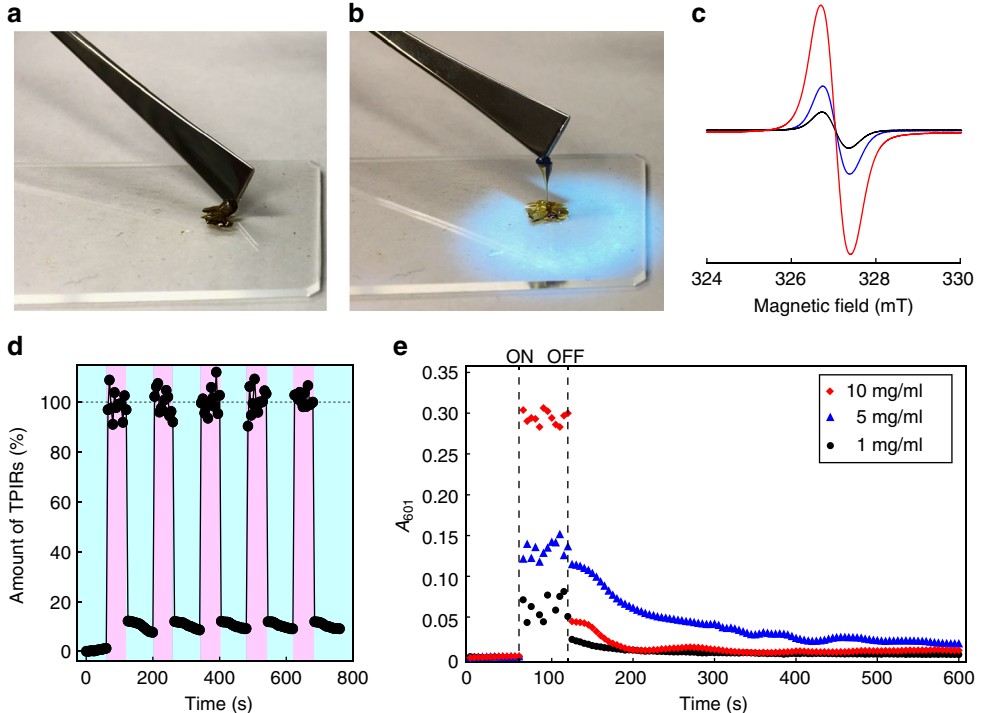

**Fig. 2** IRLNC experiment with **network-PBA_HABI** network-PBA_HABI. Photographs of **network-PBA_HABI a** before and **b** after UV irradiation. **c** ESR spectra of **network-PBA_HABI** before (*black*) and during (*red*) UV irradiation for 1 min, and 1 min after terminating UV irradiation (*blue*). **d** Time-dependent plots of the amounts of TPIR in a THF solution of **network-PBA_HABI** (10 mg/ml). The amount of TPIR was determined by recording $A_{601}$ every 5 s. The sample was irradiated with UV and not irradiated with UV during times corresponding to the *pink* and *light blue regions*, respectively. The value of $A_{601}$ before UV irradiation was defined as 0% and the averaged $A_{601}$ during UV irradiation was defined as 100%. **e** Time-dependent plots of absorbance at 601 nm for **network-PBA_HABI** in THF. Initial **network-PBA_HABI** concentration was 1 mg/ml (*black circle*), 5 mg/ml (*blue triangle*), and 10 mg/ml (*red diamond*), respectively. The starting (ON) and terminating (OFF) times of UV irradiation were indicated with dashed lines

liquid-star-shaped polymers can be toggled by external stimuli (Fig. 1a) and designed a repeatable MAT system relies on the cleavage and reformation of the imidazole–imidazole covalent bond in HABI (Fig. 1b). We first focused on the repeatable MAT of PBA (Fig. 1c), a common noncrystalline acrylate polymer with low $T_g$ that is in the liquid state when its molecular weight is low. Four-armed star-shaped PBAs with bromide end groups (**star-PBA_Br**s) were synthesized by atom transfer radical polymerization of *n*-butyl acrylate initiated from pentaerythritol tetrakis(2-bromoisobutyrate) (**1**) as the tetra-functional initiator (Supplementary Fig. 1a). Examination of the dependence of molecular weight on the state of the polymerized product revealed that an increase in the total molecular weight above 5500 Da, which corresponds to a degree of polymerization per arm (DP_n(arm)) of 8, resulted in solidification of **star-PBA_Br** (Supplementary Table 1). A compound with four lophines (**2**), which is a model small molecule for a star polymer with a DP_n(arm) of 0, was also synthesized (Supplementary Fig. 2). Compound **2** was a crystalline solid, and thus it was suggested that attaining IRLNC with PBA at least requires a DP_n(arm) between 1–7. In the present study, **star-PBA_Br** with a DP_n(arm) of 3 was selected for further study (Supplementary Table 1, entry 1). The four bromide end groups of **star-PBA_Br** were converted to azide groups by reacting with NaN$_3$ and the subsequent Huisgen reaction with an ethynyl derivative of lophine (**3**) afforded a four-armed star-shaped PBA with lophine end groups (**star-PBA_Lophine**) (Supplementary Fig. 1a). The resulting **star-PBA_Lophine** was thoroughly characterized by $^1$H NMR (Supplementary Fig. 3a), GPC (Supplementary Fig. 4), and MALDI-TOF mass spectrometry (Supplementary Fig. 5). Oxidation reaction of the lophines followed by a coupling reaction of the generated TPIRs provided a

network PBA (**network-PBA_HABI**) (Supplementary Fig. 1a). The synthesized **network-PBA_HABI** was a nonliquid material poorly soluble in common organic solvents. The $^1$H NMR spectrum of the CDCl$_3$ soluble part of **network-PBA_HABI** showed that the complex signals in the aromatic region (Supplementary Fig. 3b), which did not appear before the reaction (Supplementary Fig. 3a). These signals are due to differences in the type of imidazole–imidazole covalent bond in HABI[29, 30]. **Star-PBA_Lophine** and **network-PBA_HABI** were further subjected to $^1$H NMR analyses using DMSO-$d_6$ to determine conversion yield from TPIRs to HABIs. By comparing the integration of the –NH signal at 12.5 p.p.m. appeared before (Supplementary Fig. 6a) and after the reaction (Supplementary Fig. 6b), conversion yield from TPIRs to HABIs was determined to be ca. 60%.

When UV ($\lambda = 365$ nm) was irradiated to the produced **network-PBA_HABI** under isothermal condition at 50 °C, the nonliquid **network-PBA_HABI** (Fig. 2a) liquefied to afford a viscous material (Fig. 2b). This change was complete within 1 min (Supplementary Movie 1) and the material reverted to the nonliquid state within 3 min after terminating UV irradiation. Comparison of the ESR spectra before and after UV irradiation for 1 min directly demonstrated cleavage of the imidazole–imidazole covalent bond in HABI to produce a pair of TPIRs (Fig. 2c, *red curve*). The g-tensor was measured to be 2.003, which exactly corresponded to the reported values for TPIR[31, 32]. Comparison of the ESR spectra before and after UV irradiation also showed a 10-fold increase in the amount of TPIR after UV irradiation (Fig. 2c, *black* and *red curves*). After terminating UV irradiation for 3 min, the generated TPIRs recoupled with each other, resulting in a decrease in the relative signal intensity (Fig. 2c, *blue curve*). The presence of a small

**a**

Star-PBA$_{Lophine}$

1 mg/ml

5 mg/ml

10 mg/ml

50 mg/ml

100 mg/ml

12          14          16
Elution time (min)

**b**

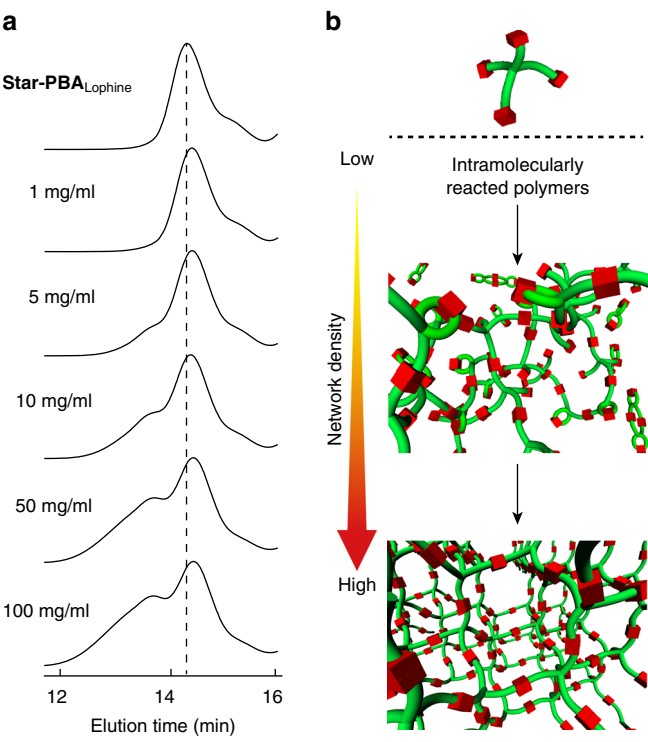

Low

Intramolecularly
reacted polymers

Network density

High

**Fig. 3** Concentration-dependent repeatable MAT. **a** GPC traces of the products after UV irradiation to THF dispersions (1–100 mg/ml) of **network-PBA$_{HABI}$**. **b** Schematic illustration of the dependence of the network density on concentration

amount of TPIRs before and after UV irradiation suggests that the retarded recoupling of TPIRs in the later stage of network formation reaction and that difficulty in diffusion for polymer chains in the network matrix. Nevertheless, the TPIRs and their recoupling reaction indeed functioned as an IRLNC system.

The mechanism of the reaction was investigated using three concentrations (1, 5, and 10 mg/ml) of **network-PBA$_{HABI}$** dispersions in THF. UV irradiation to the dispersions completely dissolved **network-PBA$_{HABI}$** in THF and the solution became blue. The blue color disappeared after terminating UV irradiation for several minutes. These color changes were repeatable (Supplementary Fig. 7) and could be quantitatively monitored by absorbance at the wavelength of 601 nm ($A_{601}$) of the TPIR molecules[33] by comparing the UV–vis spectra of **star-PBA$_{TPIR}$** and **network-PBA$_{HABI}$** (Supplementary Fig. 8). Time-dependent plots of the amount of TPIRs obtained using the **network-PBA$_{HABI}$** concentration of 10 mg/ml demonstrated repetitive increases and decreases in the amount of TPIRs by turning the UV lamp on and off and most of these changes occurred within 5 s (Fig. 2d). Although a small amount of TPIRs remained after turning the UV lamp off for 80 s, ~95% of the TPIRs recoupled within 5 s. The production of TPIRs was essentially instantaneous irrespective of concentration, but two different series of TPIR recoupling reactions were evident from the decreases in $A_{601}$ (Fig. 2e). The existence of these two trends in the decrease of absorbance at $A_{601}$ is likely due to a difference in the reaction rate between inter- and intra-molecular recoupling of TPIRs. The 10 mg/ml solution showed the apparently higher rate for the first sudden decrease in $A_{601}$ compared with the other concentrations tested (Fig. 2e). A higher concentration in general allows faster intermolecular reaction and thus the first sudden decrease in $A_{601}$ likely corresponds to intermolecular recoupling of the TPIRs. The rate of the subsequent second decreasing trend in $A_{601}$ (2–10 min) is apparently slower than the first one yet is nonetheless

remarkably faster than those of recoupling reactions of typical TPIRs (>1 h)[33, 34]. The coupling reaction of TPIRs in dilute solution is slow because the TPIR molecules are far apart. However, similar to the bridged imidazole dimers reported by Abe et al.[35], the four TPIRs in the single polymer chain likely caused accelerated intramolecular recoupling of **star-PBA$_{TPIR}$**s, resulting in a faster reaction rate than those of reported ones[33, 34]. Concentration-dependent GPC measurements were performed to test this hypothesis. Comparison of the GPC chromatograms of THF solutions containing different concentrations of **network-PBA$_{HABI}$** prepared by irradiating UV to the corresponding dispersions showed products due to the intermolecular reaction at concentrations above 5 mg/ml, and these **network-PBA$_{HABI}$**-derived fractions increased as the concentration increased (Fig. 3a). On the other hand, the peak molecular weights ($M_p$s) shifted toward lower molecular weight region compared with **star-PBA$_{lophine}$**, where the $M_p$ of **star-PBA$_{lophine}$** is indicated by a dashed line in Fig. 3a. A reduction in hydrodynamic volume is a well-known effect of the formation of intramolecularly linked polymer products, for example, cyclic and eight-shaped polymers[16]. These results support our hypothesis regarding inter- and intra-molecular reactions and higher concentrations increased the ratio of **network-PBA$_{HABI}$** (Fig. 3b).

Taken together, the results demonstrate that the desired IRLNC relying on repeatable MAT has been achieved and well-defined star polymers allowed detailed analysis of the mechanism underlying the present repeatable MAT. There are several reports describing polymers with lophine side chains[31, 36] that characterize in detail their side-chain oxidation and UV responses; however, to achieve IRLNC with such polymers is difficult because of their inherent solidity. On the other hand, the molecular weight range of **star-PBA$_{Lophine}$** and **star-PBA$_{TPIR}$** compatible with maintaining the liquid state is limited, and the temperature required to demonstrate IRLNC with bulk **network-PBA$_{HABI}$** was relatively high (vide ante). Wherein, we next focused on the IRLNC of PDMS. Four-armed star PDMSs with hydrosilane end groups (**star-PDMS$_{SiH}$**) were synthesized by organocatalytic ring-opening polymerization of hexamethylcyclotrisiloxane[37] initiated from pentaerythritol and terminated with chlorodimethylsilane (Supplementary Fig. 1b). Subsequent hydrosilylation with 4-allyloxy benzaldehyde using Karstedt's catalyst afforded star-shaped PDMSs with aldehyde end groups (**PDMS$_{Aldehyde}$s**). Star-shaped PDMSs with lophine end groups (**star-PDMS$_{lophine}$s**) were then synthesized by the reaction of **star-PDMS$_{Aldehyde}$** with benzil using Debus–Radziszewski imidazole synthesis conditions; subsequent oxidation to generate TPIRs followed by their coupling reaction afforded the corresponding network polymer (**network-PDMS$_{HABI}$**) (Supplementary Fig. 1b). The star-shaped PDMSs were designed to have core pentaerythritol-derived branch points similar to those of star-shaped PBAs to minimize the effect of the branch points on properties. Comparison of the GPC chromatograms of **star-PDMS$_{SiH}$** (Supplementary Fig. 9a), **star-PDMS$_{Aldehyde}$** (Supplementary Fig. 9b), **star-PDMS$_{Lophine}$** (Supplementary Fig. 9c) confirmed that the above-described sequential reactions did not cause degradation reactions such as cleavage of the relatively weak Si–O–C linkages. Although the molecular weight distributions of these star-shaped PDMSs were broad in comparison with those of star-shaped PBAs (Supplementary Fig. 4), the targeted **star-PDMS$_{Lophine}$** was well-characterized by $^1$H NMR (Supplementary Fig. 10a). Under the range of experimental conditions tested, the inherent liquid nature of PDMS and relatively long polymer chains prevented solidification of **star-PDMS$_{Lophine}$** despite the terminal four lophines (Supplementary Table 2). The subsequent network formation by oxidizing the terminal lophines afforded nonliquid **network-PDMS$_{HABI}$**. The $^1$H NMR spectrum

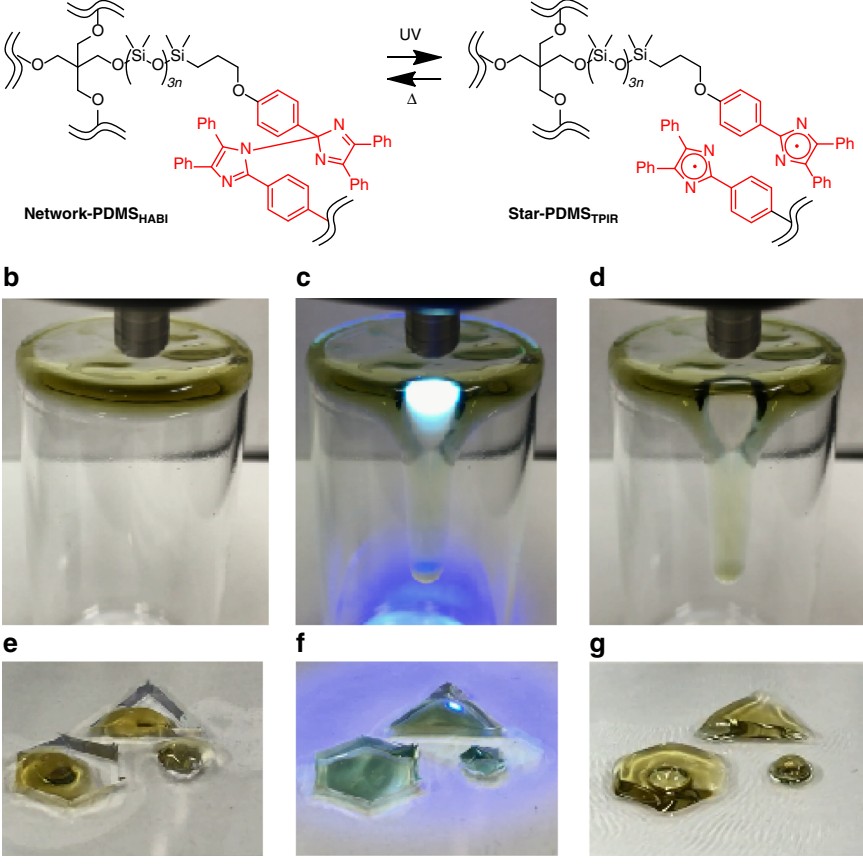

**Fig. 4** IRLNC experiment with **network-PDMS_HABI**. **a** Repeatable MAT between **network-PDMS_HABI** and **star-PDMS_TPIR** based on UV-triggered bond cleavage and reformation. Photographs of **network-PDMS_HABI** on the vial **b** before and **c** during UV irradiation for 3 min 43 s and **d** immediately after terminating UV irradiation. Photographs of **network-PDMS_HABI** on the templates **e** before and **f** during UV irradiation for 9 min 55 s, and **g** after terminating UV irradiation and removing the template plastic sheet

of CDCl$_3$ soluble part of **network-PDMS_HABI** showed changes of signals at the aromatic region (Supplementary Fig. 10b) similar to those recognized for **network-PBA_HABI**. The conversion from TPIRs to HABIs was also determined by comparing $^1$H NMR spectra of **star-PDMS_Lophine** (Supplementary Fig. 11a) and acetone-$d_6$ soluble part of **network-PDMS_HABI** (Supplementary Fig. 11b). The conversion yield was calculated to be ca. 70% from these spectra. The GPC chromatogram of the THF soluble part of **network-PDMS_HABI** showed a shift of the trace toward a higher molecular weight region (Supplementary Fig. 9d) compared with that of **star-PDMS_Lophine** (Supplementary Fig. 9c), demonstrating that the TPIRs in the generated **star-PDMS_TPIR** reacted intermolecularly with each other. Importantly, no lower-molecular-weight fractions were observed in the GPC chromatogram of **network-PDMS_HABI**, suggesting that the Si–O–C linkages near the junction points are stable under the reaction conditions tested and even under the existence of TPIRs.

We further attempted IRLNC based on repeatable MAT between **network-PDMS_HABI** and **star-PDMS_TPIR** (Fig. 4a) by UV irradiation to nonliquid **network-PDMS_HABI** deposited on the surface of a vial (Fig. 4b). Significantly, room temperature (25 °C) IRLNC was achieved (Supplementary Movie 2). The **network-PDMS_HABI** only at the area exposed to UV liquefied and flowed down (Fig. 4c). The slight blue coloration and liquefaction of the material suggested production of **star-PDMS_TPIR**s. The liquid state due to **star-PDMS_TPIR** was readily reverted by terminating UV irradiation and nonliquid **network-PDMS_HABI** reformed on the glass surface (Fig. 4d). Remarkably,

the present IRLNC completed within a few minutes (Supplementary Movie 2). Furthermore, we have succeeded in applying **network-PDMS_HABI** to a photochemically metamorphosing material (Supplementary Movie 3). Thus, a plastic sheet with circle, triangle, and hexagonal shaped templates was fixed on a slide glass and the **network-PDMS_HABI** was deposited on a surface of these templates (Fig. 4e). The nonliquid **network-PDMS_HABI** on the templates liquefied upon UV irradiation and the liquefied material, that is, **star-PDMS_TPIR** spread within the templates (Fig. 4f). After terminating UV irradiation and removing the plastic sheet with the templates, the reverted **network-PDMS_HABI**s with circle, triangle, and hexagonal shapes were formed on the slide glass (Fig. 4g). This demonstrates that the present molecular system has a potential for the repeatable photochemical molding technology[14] with the different mechanism from conventional ones.

**Rheological aspects of photo-triggered solvent-free IRLNC.** Dynamic mechanical analysis (DMA) was performed to directly reveal the change of storage modulus ($G'$) and loss modulus ($G''$) in the present IRLNC system. Time-dependent plots of $G'$ (Fig. 5a) and $G''$ (Fig. 5b) of **network-PDMS_HABI** demonstrated their repetitive decreases and increases by turning the UV lamp on and off (Fig. 5). The measurements were repeated three times (1st: *black circle*, 2nd: *blue square*, and 3rd: *red triangle*) and all of the measurements showed similar trends, demonstrating the reversibility of the present liquid–nonliquid conversions. More importantly, the temperature change of the material monitored

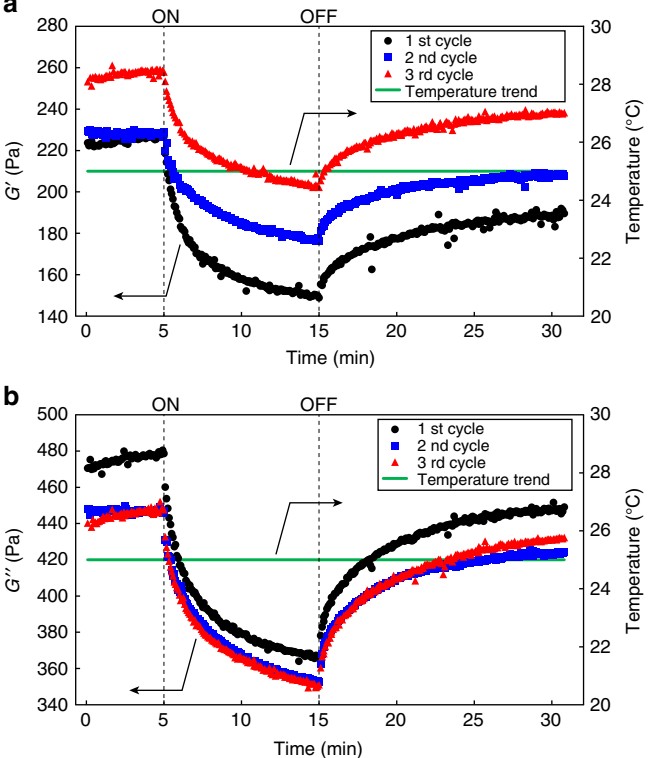

**Fig. 5** Rheological properties upon repetitive UV irradiations. Time-dependent plots of **a** $G'$ and **b** $G''$ for **network-PDMS**$_{HABI}$ upon repeating UV irradiation cycles (1st cycle: *black circle*, 2nd cycle: *blue square*, and 3rd cycle: *red triangle*) with monitoring temperature (*green line*). The starting (ON) and terminating (OFF) times of UV irradiation were indicated with *dashed lines*

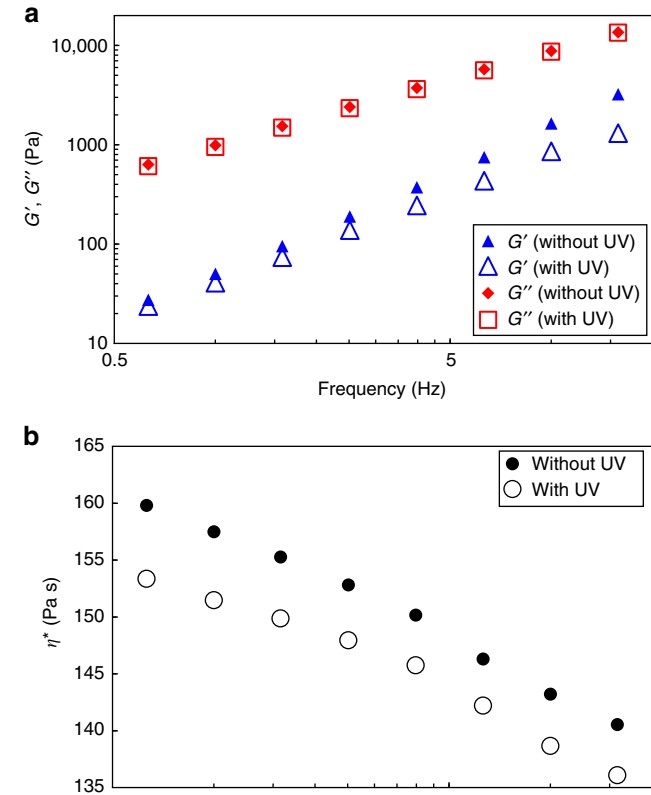

**Fig. 6** Frequency dependence on rheological properties with and without UV irradiation. **a** Frequency-dependent plots of $G'$ and $G''$ for **network-PDMS**$_{HABI}$ ($G'$ without UV: *blue triangle*, $G'$ with UV: *blue open triangle*, $G''$ without UV: *red diamond*, $G''$ with UV: *red open square*). **b** Frequency-dependent plots of $\eta^*$ without (*black circle*) and with (*black open circle*) UV irradiation

during the measurements was negligible (Fig. 5, *green line*). DMA analysis of **network-PBA**$_{HABI}$ under irradiating UV at 50 °C likewise showed repetitive decreases and increases of both of $G'$ (Supplementary Fig. 12a) and $G''$ (Supplementary Fig. 12b), indicating that the present repeatable MAT-based IRLNC has a certain level of generality.

Meanwhile, DMA analysis of **network-PDMS**$_{HABI}$ showed higher $G''$ than $G'$ irrespective of UV irradiation, which is more typical for liquid materials. From this characteristic and the actual nonliquid appearance of **network-PDMS**$_{HABI}$ without UV irradiation (Supplementary Movie 2), two reasons can be considered for the present IRLNC. One is high-complex viscosity ($\eta^*$) of **network-PDMS**$_{HABI}$ enough to avoid flow of the material. The other is the existence of crossovers with the frequency-dependent curves of $G'$ and $G''$. In fact, various polymer materials show frequency dependence of $G'$ and $G''$ and whether we should define the materials as liquid or nonliquid relies on a time scale for observing the phenomenon as recognized in the pitch drop experiment started from 1927 and still continues[38]. Moreover, Sheiko and coworkers reported unique dependence of $G'$ and $G''$ on frequency with solvent-free supersoft and superelastic bottlebrushes composed either of PBA or PDMS[39, 40]. Thus, we further examined frequency dependence of $G'$ and $G''$ with **network-PDMS**$_{HABI}$. Based on frequency dependent DMA measurements for **network-PDMS**$_{HABI}$, it was revealed that the plots of $G''$ were always greater than those of $G'$ irrespective of UV irradiation in the experiment range (Fig. 6a) and $\eta^*$s at the corresponding frequencies were ~150 Pa s (Fig. 6b). Given that $\eta^*$s of commercial linear PDMS melts were in the range of 1.0–1.5 Pa s at a similar frequency range[41], the measured $\eta^*$s for

**network-PDMS**$_{HABI}$ were roughly hundredfold greater than those of such PDMS melts. The decreased amounts of $G'$ and $G''$ at the frequency around 3 Hz did not correspond to that appeared in the temperature-dependent measurement (Fig. 5) likely due to the difference in thickness of the samples. However, the frequency dependent DMA measurements also revealed that the decrease of $G'$ upon UV irradiation was more prominent than that of $G''$ (Fig. 6a).

## Discussion

IRLNC of polymeric materials relying on repeatable MAT between star and network architectures was demonstrated. The key for the present repeatable MAT was cleavage and reformation of the imidazole–imidazole covalent bond in HABI, which enabled prompt IRLNC in response to UV. The mechanism for the present IRLNC was further examined by DMA studies. Generally, the existence of network structure is primarily responsible for exhibiting elasticity and **network-PDMS**$_{HABI}$ showed greater shift of $G'$ than $G''$ upon UV irradiation. Therefore, the decrease of $\eta^*$ or $G'$ by UV irradiation presumably caused the photo-triggered solvent-free IRLNC. Moreover, IRLNC methodology is, in principle, general as long as the parent star-shaped polymers are liquid. Therefore, the described repeatable MAT-based IRLNC and photo-triggered repeatable metamorphosis of polymers open the door to soft materials for use in various fields such as recyclable and deformable structural

materials and adhesives, including human body-related applications.

## Methods

**Synthesis of network-PBA$_{HABI}$.** Into a flask was prepared an ethyl acetate (10 ml) solution of **star-PBA$_{Lophine}$** (170 mg, 0.174 mmol for end groups), and an aqueous solution (10 ml) containing potassium ferricyanide (2.86 g, 8.7 mmol) and potassium hydroxide (488 mg, 8.7 mmol) was added. The mixture was stirred at room temperature for 60 min. The aqueous layer was separated and extracted with ethyl acetate. The combined ethyl acetate phases were washed with water, dried over MgSO$_4$, and evaporated. After drying under reduced pressure, an elastic solid poorly soluble to organic solvents (**network-PBA$_{HABI}$**) was obtained. The yield was 142 mg. $^1$H NMR (500 MHz, CDCl$_3$) $\delta$ p.p.m. 0.88 (s, –CH$_2$CH$_3$), 1.08 (s, –C (CH$_3$)$_2$–), 1.21–2.65 (m, –CH$_2$CH(COO$^n$Bu)–, –CH$_2$CH$_2$CH$_3$), 4.05 (m, –COOCH$_2$–), 5.12 (m, triazole–CH$_2$OAr), 5.38 (m, –CH$_2$CH(COO$^n$Bu)–triazole), 6.85–7.93 (–CH$_2$OAr*H*). $M_n$ (RI) = 4300, $M_p$ (RI) = 7200, $M_w/M_n$ = 2.34.

**Synthesis of network-PDMS$_{HABI}$.** Into a flask was prepared a hexane (10 ml) solution of **star-PDMS$_{Lophine}$** (1.35 g, 0.45 mmol for end groups), and an aqueous solution (30 ml) containing potassium ferricyanide (7.41 g, 22.5 mmol) and potassium hydroxide (1.26 mg, 22.5 mmol) was added. The mixture was stirred at room temperature for 60 min. The aqueous layer was separated and extracted with hexane. The combined hexane phases were washed with water, dried over MgSO$_4$, and evaporated. After drying under reduced pressure, an elastic solid poorly soluble to organic solvents (**network-PDMS$_{HABI}$**) was obtained. The yield was 1.32 g. $^1$H NMR (500 MHz, CDCl$_3$) $\delta$ p.p.m. 0.03–0.53 (s, –CH$_3$), 0.61 (s, –CH$_2$CH$_2$CH$_2$OAr), 1.75 (s, –CH$_2$CH$_2$CH$_2$OAr), 3.45–3.89 (m, –CCH$_2$O–, –CH$_2$CH$_2$CH$_2$OAr), 7.00–8.22 (m, Ar*H*). $M_n$ (RI) = 6100, $M_p$ (RI) = 11200, $M_w/M_n$ = 2.34.

**UV irradiation experiments**. UV irradiation experiments were performed using a Hamamatsu LIGHTNING CURE LC8 L9588 model with a Hamamatsu A10014-35-0110 light guide. A Hamamatsu A9616-07 filter was used for efficiently transmitting only the UV light around 365 nm (150 mW/cm$^2$). UV irradiation to the bulk **network-PBA$_{HABI}$** was performed on the slide glass heated at 50 °C with hot plate. For concentration-dependent experiments, UV was directly irradiated to the quartz cells containing THF dispersions of **network-PBA$_{HABI}$** (1, 5, 10, 50, and 100 mg/ml). Further, UV irradiation to the bulk **network-PDMS$_{HABI}$** inside the vial was performed at 25 °C.

**Rheological analyses**. Rheological analyses under the irradiation of UV (365 nm, 150 mW/cm$^2$) were performed on an Anton Paar MCR 302 rheometer equipped with a UV curing system and a Pertier temperature control device. Time-dependent analyses of storage modulus ($G'$) and loss modulus ($G''$) were conducted using a parallel plate with a diameter of 12 mm at the frequency of 3 Hz for the samples with the thickness of 0.5 mm. Temperatures of the parallel plate were set to 50 °C for **network-PBA$_{HABI}$** and to 25 °C for **network-PDMS$_{HABI}$** and were monitored during the measurements. Each time-dependent measurement was conducted for 30 min and then the sample was held sufficient time to reach equilibrium state before starting the following measurements. For frequency-dependent $G'$ and $G''$ measurements, the **network-PDMS$_{HABI}$** sample with the thickness of 0.12 mm was employed.

**Data availability**. The data that support the findings of this study are available from the corresponding authors upon reasonable request.

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

## Acknowledgements

We are grateful to Mr. Yuichi Shinozaki and Mr. Yasuhito Kajita at Anton Paar Japan K.K. for our access to the DMA apparatus and kind support for the analyses. This work was supported partly by JSPS KAKENHI (Grant Numbers 16K14074 S.H. and 16H04032 T.T.) and Izumi Science and Technology Foundation (S.H.).

## Author contributions

S.H. designed the work and performed the experiments. S.H. and T.T. wrote the manuscript.

## Additional information

**Competing interests:** The authors declare no competing financial interests.

