## [Peer Review File · Nature Communications]

Reviewers' comments:

Reviewer #1 (Remarks to the Author):

The authors investigated the synthesis, characterization, and reversible light-induced solidification and liquefaction in molecular systems between star and network polymers utilizing cleavage and formation of a covalent bond in HABI. Authors reported that the formation of the network polymer solidifies the sample. In addition, breaking the network structure to produce the TPIRs by irradiation of UV light, the sample turns to liquid state. Generation of TPIRs were confirmed by ESR and the formation of network polymers were characterized by GPC and NMR. The phenomena of isothermal reversible solidification and liquefaction (IRSL) is quite interesting and the strategy of the molecular architecture is intriguing. However, at this moment, the experimental data is not enough, the background of the study is not provided fairly and the manuscript seems to be somewhat misleading. Therefore, I strongly recommend the authors to revise the manuscript and resubmit. Please refer to the following comments/concerns.

1) The characterization of the polymer is not sufficiently reported. Provide DSC of the polymers and the change of the melting temperature before and after the photoirradiation. Also, rheological parameters should be provided if authors wish to claim the sample is in solid or liquid phase. In Movie 3, the initial state of the "solid" sample seems to be viscose fluid, as the initial sample seems to be spreading between two transparent sheets due to the capillary effect.

2) The effect of temperature rise by photoirradiation is not discussed in this manuscript. When irradiated, the temperature of the sample should increase. In this manuscript, I doubt the melting is taking place because the sample is heated to the melting temperature. The heating effect has been discussed in the related previous reports (especially see: *Org. Lett.* 2014, 16, 5012).

3) In the second paragraph in the main text, authors describe the light-induced melting of azobenzene-containing molecules. However, I found several descriptions are not correctly presented and rather misleading. Also, there are some previous works to be cited.

3-1. Line 31, "...in response to UV4-7," Some works are missing. Refer to: *ChemCommun* 2011, 47, 1770. (the first report), *Chem. Eur. J.* 2013, 19, 17291., *Org. Lett.* 2014, 16, 5012. (already mentioned in 2), *J. Photopolym. Sci. Tech.* 2016, 29, 149.

3-2. Line 31-38, "but this approach can work only ... whole polymer chain relaxation." The above sentences are not correct and misleading. Simple small molecules exhibit the melting upon photoirradiation as reported in *CrystEngComm*, 2016, 18, 7225. and *Nat. Commun.* 2015, 6, 7310. In addition, recent reports show that several azobenzene polymers exhibit the melting upon irradiation as reported in *J. Adhes.* 2016, DOI: 10.1080/00218464.2016.1219255. and *Nat. Chem.* 2016, DOI: 10.1038/nchem.2625. Indeed, azobenzene molecular systems with wide range of chemical structure show the light-induced melting. Therefore, this paragraph should be rewritten.

4) line 50, the definition of the "RMAT" is not clear. Provide the definition, or give a reference if it was previously defined.

5) line 195, "... a novel repeatable photochemical molding technology." However, a similar concept has just been published in *Nat. Chem.* 2016, DOI: 10.1038/nchem.2625. (the paper mentioned in 3-2).

6) line 95, "Figure 1c" should be "Figure 2c".

7) Table S1, there are two "entry 1". What is the difference between these two entries?

Reviewer #2 (Remarks to the Author):

Nature Communication Review Report

Dear Editor,

It is a pleasure for me to join the peer review of the paper entitled: "Photo-triggered solvent-free metamorphosis of polymeric materials", which was delivered to Nature Communication. This paper

provides a new idea of controlling the transformation between solidification and liquefaction by using the photo induced cleavage of HABI in nonlinear polymers, which is interesting to broad readers. Generally, the experimental results support their claims. However, in my opinion, there are still some points needed to be improved for more details and discussions. We have prepared our review report on this paper and presented as below.

Q1. In Figure 2(c), EPR is used to monitor the photo cleavage of the polymers and we could see the EPR signal increases indeed, which could be well explained by the photochromic nature of HABIs. However, it should be noted that HABI exhibits no EPR signals before light irradiation, while a strong and clear EPR signal was observed (Figure 2c, black line) before UV irradiation. On the other hand, it seems as if the EPR signals could not be totally recovered or to the initial state level according to the data obtained by the authors. These characters indicates that the reversibility of this system might not be good enough. Could these defects be improved or solved in this system through some kinds of chemical or modifications? The discussion about this point will be valuable.

Q2. It is confusing for me about the calculation of the amount of TPIRs proposed by the authors in Figure 2(d). I mean how are these values obtained? Probably the authors set the apparent TPIR content to be 100% at the UV-equilibration and 0% at the beginning. As the authors have claimed that the apparent TPIRs content is almost 0% before irradiation in Figure (d) (the real value seems not 0% according to the results of EPR (Figure (c))). I suggest the authors should check this issue or provide a clearer description on the amount of TPIRs in this polymer system.

Q3. In Figure 2(e), the red line (+square, 10 mg/ml) represents the fading kinetics of network-PBA-HABI and the results is confusing for me. Clearly, with higher concentration of network-PBA-HABI, the A601 will be stronger after UV irradiation in the initial state. However, a sharp decrease of A601 value in 10 mg/ml solution was observed, which is quite different from the sample 1 mg/ml and 5 mg/ml. The authors have proposed two kinds of processes involved in it, which are inter- and intramolecular interactions controlled by concentration. In fact, the change of absorbance upon UV irradiation in 10 mg/ml sample is almost instantaneous. From the current data provided by the authors, we could hardly get much useful information or indication of fast-fading events. Thus, we suggest the authors to repeat this measurement or conduct transient UV-vis absorption to illustrate this hypothesis. Based on these data, we should expect the fading kinetics with two different stages. Meanwhile, a series of concentration between 5 and 10 mg/ml should also be investigated to obtain the critical polymer concentration of this unusual behavior.

Q4. In Figure 2(e), different concentration of this polymer solution were prepared to investigate the fading kinetics of the colored species. However, these fading kinetics spectra are rather rough with clear vibrations and we suggest these results should be improved to increase the quality of this work.

Q5. The reversibility of these HABI-polymers are only investigated in solution state in this paper. The main claim of this paper was focused on the transformation of these polymers in solid state to "melted" state. Thus we suggest more properties like photochromism, fatigue resistance and fading kinetics should be investigated in solid or in polymer matrix to emphasize the advantage of this system.

Q6. The concentration effect, which was proposed by the authors, was used to illustrate the reconstruction of the HABI-polymers in Figure 3. The GPC spectra should be the same either by diluting or concentrating this freshly prepared polymer solution without UV irradiation according to the hypothesis proposed by the authors. I feel that the star-PBAHABI solution before UV irradiation would be better to be used as standard to exhibit the reconstruction happening after UV irradiation.

Q7. In Figure S7, the classification of -NH (imidazole NH, marked as f in the supporting information) to a small peak with chemical shift around 9 should be carefully checked. According to

previous investigations and papers, this -NH signal is very typical for triphenylimidazoles (TPIs) whose chemical shift value is generally around 12~13. (Lots of recent publication papers could be found to support this) As far as I am concerned, this value could hardly be located around 9. Meanwhile, this -NH signal is also used as an indicator to determine the conversion yield from TPIs to HABIs, especially for polymers. However, in this paper, the NMR results of either star-PDMSLophine or network-PDMSHABI are tested by using CDCl₃ as solvent and the spectra were all presented with spectra range round 0-11. Meanwhile, it should be noted that the -NH signal from the imidazole unit could be covered by the solvent CDCl₃ and we suggest d-DMSO is the best solvent. On the other hand, it suggests that the integrated areas of every selected peaks be provided. Although this issue seems very tiny, we suggest the authors to check it carefully on this point.

Q8. In Figure S2, the caption "SEC trace of star-PBALophine" better be corrected to "GPC trace of star-PBALophine" to be accorded with the text.

Q9. In Figure S3, the MALDI-TOF spectrum of star-PBALophine was presented, not the corresponding of the final product star-PBAHABI. The identification and presentation of more information of the final products are suggested.

Other suggestions:

It is found that azo-benzene contained polymers, which exhibit photo induced T_g transformation between liquid and solid state induced by cis-trans exchange of azo-benzene has been reported by Wu in a recent paper (Title: Photoswitching of glass transition temperatures of azobenzene-containing polymers induces reversible solid-to-liquid transitions, Nature Chemistry (2016) doi:10.1038/nchem.2625). These results covers the new developments of azo-benzene based materials. We suggest this paper should be cited.

Based on all these comments above, we suggest the authors to revise their manuscript before a final decision is reached.

Reviewer #3 (Remarks to the Author):

This paper describes the photoinduced liquefaction of network polymer. The author demonstrated the interesting phenomena that the polymer liquefied under light exposure, and mentioned that it is due to bond cleavage of hexaarylbiimidazole. This concept is new.

Therefore it is necessary to prove experimentally that the mechanism is mainly working for this phenomena. However, there is an unclear point in this regard.

The authors have analyzed solution or dispersion state of the network polymers by means of spectroscopic measurement, GPC measurement and NMR measurement. Because the network polymer is usually insoluble, it indicates containing of a soluble component composed of incomplete-network polymer. In fact the authors discuss the intramolecular dimerization in the manuscript. If the soluble component is a main part of the polymer, the influence of thermal plasticization on liquidity should be also taken into account. Relatively strong light source (like 150mwcm⁻²) used here is often increase the temperature.

Since control of the liquid-solid state of the substance is a basic part related to materials science, a lot of people will have interest in this subject. The concept is different from previous cases.

Therefore, if the authors make it clear in their response that the influence of the temperature rise by light irradiation is negligible, I recommend the publication. In this case, it is necessary to reconsider the description at line 29-31, because there are recently reported on polymer systems (Nature Chemistry (2016) doi:10.1038/nchem.2625, Journal of Adhesion (2016) doi.org/10.1080/00218464.2016.1219255)

Point-to-Point Answers to the Referees' Comments.

First of all, we thank the reviewers very much for the valuable compliments and suggestions. According to the comments, we have conducted additional experiments and elaborated our manuscript.

For Reviewer 1:

Comment #1

The characterization of the polymer is not sufficiently reported. Provide DSC of the polymers and the change of the melting temperature before and after the photoirradiation. Also, rheological parameters should be provided if authors wish to claim the sample is in solid or liquid phase. In Movie 3, the initial state of the "solid" sample seems to be viscose fluid, as the initial sample seems to be spreading between two transparent sheets due to the capillary effect.

Response

First, melting temperatures (T_m s) of star- and network-poly(butyl acrylate) (PBA) are not measurable because PBA is noncrystalline (amorphous) polymer and thus does not show T_m . On the other hand, comparison of thermal properties before and after the photoirradiation is difficult because the product after UV irradiation, i.e., star-PBA_{TPIR} soon react each other to again form network-PBA_{HABI}. Likewise, comparison of DSC results between network-PDMS_{HABI} and star-PDMS_{TPIR} was difficult. Second, measuring rheological parameter under the irradiation of UV was possible by using Anton Paar MCR 702 equipped with a UV irradiation option and a temperature controller, where the measurement temperature was set to 25 °C. Changes of G' and G'' with and without irradiating UV was discussed in the revised manuscript. From DMA results, the state of network-PDMS_{HABI} should be expressed as rather "nonliquid" than "solid". In the revised manuscript, the description "solid" has been replaced.

Comment #2

The effect of temperature rise by photoirradiation is not discussed in this manuscript. When irradiated, the temperature of the sample should increase. In this manuscript, I doubt the melting is taking place because the sample is heated to the melting temperature. The heating effect has been discussed in the related previous reports (especially see: Org. Lett. 2014, 16, 5012).

Response

As mentioned in the response for the comment #1, we conducted DMA measurements using Anton Paar MCR 702 equipped with a UV irradiation option and a temperature controller, where the measurement temperature was set to 25 °C. Based on DMA, UV-triggered liquefaction and loss of liquidity by terminating UV was clearly confirmed and change of the temperature under the irradiation of UV was negligible under the irradiation condition. The related discussion has been

added to the revised manuscript.

Comment #3

In the second paragraph in the main text, authors describe the light-induced melting of azobenzene-containing molecules. However, I found several descriptions are not correctly presented and rather misleading. Also, there are some previous works to be cited.

3-1. Line 31, "...in response to UV4-7," Some works are missing. Refer to: ChemCommun 2011, 47, 1770. (the first report), Chem. Eur. J. 2013, 19, 17291., Org. Lett. 2014, 16, 5012. (already mentioned in 2), J. Photopolym. Sci. Tech. 2016, 29, 149.

3-2. Line 31-38, "but this approach can work only ... whole polymer chain relaxation." The above sentences are not correct and misleading. Simple small molecules exhibit the melting upon photoirradiation as reported in CrystEngComm, 2016, 18, 7225. and Nat. Commun. 2015, 6, 7310. In addition, recent reports show that several azobenzene polymers exhibit the melting upon irradiation as reported in J. Adhes. 2016, DOI: 10.1080/00218464.2016.1219255. and Nat. Chem. 2016, DOI: 10.1038/nchem.2625. Indeed, azobenzene molecular systems with wide range of chemical structure show the light-induced melting. Therefore, this paragraph should be rewritten.

Response

We are grateful to the careful reading of our manuscript. According to the suggestion, the related description in the main text has been revised and the references have been cited in the revised manuscript.

Comment #4

line 50, the definition of the "RMAT" is not clear. Provide the definition, or give a reference if it was previously defined.

Response

Transformation of macromolecular architectures is one of the well-studied issues in the field of synthetic polymer chemistry. Therefore, we believe that the description "macromolecular architectural transformation (MAT)" have no problem. Some related references have been cited and only "MAT" was used as abbreviated term and "repeatable MAT" was used instead of "RMAT" to attain better readability in the revised manuscript.

Comment #5

line 195, "... a novel repeatable photochemical molding technology." However, a similar concept has just been published in Nat. Chem. 2016, DOI: 10.1038/nchem.2625. (the paper mentioned in 3-2).

Response

The reference has been cited in the revised manuscript and the main text has been revised.

Comment #6

line 95, "Figure 1c" should be "Figure 2c".

Response

The description has been modified in the revised manuscript.

Comment #7

Table S1, there are two "entry 1". What is the difference between these two entries?

Response

The differences between these two entries are polymerization time and monomer conversion. Since, monomer conversion in entry 1 was 80%, prolonged polymerization (5 h) was conducted in entry 2. The appearance of two "entry 1" in Table S1 is a simple error, the numbering of Table S1 has been revised and polymerization time of entry 2 was described in the table caption.

For Reviewer 2:**Comment #1**

In Figure 2(c), EPR is used to monitor the photo cleavage of the polymers and we could see the EPR signal increases indeed, which could be well explained by the photochromic nature of HABIs. However, it should be noted that HABI exhibits no EPR signals before light irradiation, while a strong and clear EPR signal was observed (Figure 2c, black line) before UV irradiation. On the other hand, it seems as if the EPR signals could not be totally recovered or to the initial state level according to the data obtained by the authors. These characters indicates that the reversibility of this system might not be good enough. Could these defects be improved or solved in this system through some kinds of chemical or modifications? The discussion about this point will be valuable.

Response

The remained EPR signals after the UV irradiation indicates that delayed or limited dimerization reaction likely because of difficulty in diffusing polymer chains in the formed network as discussed in the manuscript. Although change in the polymer chain length, to some extent, have a possibility to overcome this problem, screening an optimum molecular weight is beyond the scope of present study.

Comment #2

It is confusing for me about the calculation of the amount of TPIRs proposed by the authors in Figure 2(d). I mean how are these values obtained? Probably the authors set the apparent TPIR content to be 100% at the UV-equilibration and 0% at the beginning. As the authors have claimed that the apparent TPIRs content is almost 0% before irradiation in Figure (d) (the real value seems

not 0% according to the results of EPR (Figure (c)). I suggest the authors should check this issue or provide a clearer description on the amount of TPIRs in this polymer system.

Response

The absorbance at 601 nm (A_{601}) before UV irradiation was defined as 0% and averaged A_{601} during UV irradiation was defined as 100%. The definition was described in the caption of Figure 2d in the revised manuscript.

Comment #3

In Figure 2(e), the red line (+square, 10 mg/ml) represents the fading kinetics of network-PBA-HABI and the results is confusing for me. Clearly, with higher concentration of network-PBA-HABI, the A_{601} will be stronger after UV irradiation in the initial state. However, a sharp decrease of A_{601} value in 10 mg/ml solution was observed, which is quite different from the sample 1 mg/ml and 5 mg/ml. The authors have proposed two kinds of processes involved in it, which are inter- and intramolecular interactions controlled by concentration. In fact, the change of absorbance upon UV irradiation in 10 mg/ml sample is almost instantaneous. From the current data provided by the authors, we could hardly get much useful information or indication of fast-fading events. Thus, we suggest the authors to repeat this measurement or conduct transient UV-vis absorption to illustrate this hypothesis. Based on these data, we should expect the fading kinetics with two different stages. Meanwhile, a series of concentration between 5 and 10 mg/ml should also be investigated to obtain the critical polymer concentration of this unusual behavior.

Response

GPC is more powerful tool for analyzing inter- and intra-molecular reactions and thus we have presented the GPC traces of the products after UV irradiation (Figure 3). Indeed, progress of inter- and intra-molecular reactions have been clearly demonstrated from the traces and the exclusive and almost intramolecular reactions were observed with 1 and 5 mg/mL solutions, respectively. As pointed in the comment, we could hardly get much useful information of the kinetic aspects of intra- and inter-molecular reactions from UV-vis absorption measurements.

Although an elucidation of a detailed mechanism is beyond the scope of the present manuscript, we are currently analyzing it. As written in the pioneering study (*Macromolecules* **1989**, 22, 3356-3361) for analyzing gel network by small angle x-ray scattering analysis (SAXS) and recent studies (e.g., *Macromolecules*, ASAP, DOI: 10.1021/acs.macromol.7b00528, *Macromolecules* **2009**, 42, 1344-1351, and *Macromolecules* **2009**, 42, 6245-6252) for analyzing them by small angle neutron scattering analysis (SANS), the kinetic study of the present reaction system can presumably be detectable via time-resolved synchrotron-irradiation SAXS (SR-SAXS) analysis. Although we have observed concentration dependence with SR-SAXS results, further detailed analyses will be required and it is very tough work because of a beam time limitation in utilizing synchrotron-irradiation

facilities. Therefore, we believe that such studies is quite different from the present one and detailed analysis for the mechanism will be reported near future.

Comment #4

In Figure 2(e), different concentration of this polymer solution were prepared to investigate the fading kinetics of the colored species. However, these fading kinetics spectra are rather rough with clear vibrations and we suggest these results should be improved to increase the quality of this work.

Response

The A_{601} during UV irradiation was always not flat likely but somewhat noisy especially with low concentrations (1 and 5 mg/mL) as depicted in Figure 2e, likely because of scattering from the quartz cell. Nevertheless, we believe that the presented data support the results.

Comment #5

The reversibility of these HABI-polymers are only investigated in solution state in this paper. The main claim of this paper was focused on the transformation of these polymers in solid state to “melted” state. Thus we suggest more properties like photochromism, fatigue resistance and fading kinetics should be investigated in solid or in polymer matrix to emphasize the advantage of this system.

Response

In the revised manuscript, results of DMA under the irradiation of UV have been presented. The data directly demonstrate that the reversibility of these HABI-polymers in solvent-free conditions. We believe that these data support the main claim of this paper. Investigation of other suggested properties is beyond the scope of the present study.

Comment #6

The concentration effect, which was proposed by the authors, was used to illustrate the reconstruction of the HABI-polymers in Figure 3. The GPC spectra should be the same either by diluting or concentrating this freshly prepared polymer solution without UV irradiation according to the hypothesis proposed by the authors. I feel that the star-PBA_{HABI} solution before UV irradiation would be better to be used as standard to exhibit the reconstruction happening after UV irradiation.

Response

We did not synthesize star-PBA_{HABI} and star-PDMS_{HABI}, but synthesized network-PBA_{HABI} and network-PDMS_{HABI}. The possible smallest architecture with HABI-containing polymers is 8-shaped architecture as described. Moreover, star-PBA_{HABI} and star-PDMS_{HABI} cannot exist because the triphenyl imidazoryl radicals (TPIRs) generated at the ends of star-shaped polymers, i.e., star-PBA_{TPIR} and star-PDMS_{TPIR} readily reacts each other to form intermolecularly reacted products

with non-star shaped architectures. Also, isolation of star-PBA_{TPIR} and star-PDMS_{TPIR} and measuring GPC of them are difficult because these polymers are radical species. Alternatively we presented the GPC data of the THF soluble part of network-PBA_{HABI} and network-PDMS_{HABI}, and thus believe that the above concern has already been solved.

Comment #7

In Figure S7, the classification of –NH (imidazole NH, marked as f in the supporting information) to a small peak with chemical shift around 9 should be carefully checked. According to previous investigations and papers, this -NH signal is very typical for triphenylimidazoles (TPIs) whose chemical shift value is generally around 12~13. (Lots of recent publication papers could be found to support this) As far as I am concerned, this value could hardly be located around 9. Meanwhile, this -NH signal is also used as an indicator to determine the conversion yield from TPIs to HABIs, especially for polymers. However, in this paper, the NMR results of either star-PDMS_{Lophine} or network-PDMS_{HABI} are tested by using CDCl₃ as solvent and the spectra were all presented with spectra range round 0-11. Meanwhile, it should be noted that the –NH signal from the imidazole unit could be covered by the solvent CDCl₃ and we suggest d-DMSO is the best solvent. On the other hand, it suggests that the integrated areas of every selected peaks be provided. Although this issue seems very tiny, we suggest the authors to check it carefully on this point.

Response

We are grateful to the comment. According to the suggestion, we attempted ¹H NMR measurements of star-PDMS_{Lophine} and network-PDMS_{HABI} using DMSO-d₆, however, we could not obtain spectra because PDMS is insoluble in DMSO. Generally, PDMS is insoluble in DMSO, MeOH, and other polar solvents but soluble, for example, in CHCl₃ and hexane and partially in acetone. We thus screened deuterated solvents for measuring ¹H NMR and found acetone-d₆ suitable for detecting –NH signal. Likewise, DMSO-d₆ was suitable for detecting –NH signal with star-PBA_{Lophine} and network-PBA_{HABI}. In the revised manuscript, ¹H NMR spectra using acetone-d₆ for star-PDMS_{Lophine} and network-PDMS_{HABI} and DMSO-d₆ for star-PBA_{Lophine} and network-PBA_{HABI} have been presented in Figures S4 and S9, respectively. In addition, a related discussion have been updated.

Comment #8

In Figure S2, the caption “SEC trace of star-PBALophine” better be corrected to “GPC trace of star-PBALophine” to be accorded with the text.

Response

The caption of Figure S2 has been revised.

Comment #9

In Figure S3, the MALDI-TOF spectrum of star-PBALophine was presented, not the corresponding of the final product star-PBAHABI. The identification and presentation of more information of the final products are suggested.

Response

Final products we can measure MALDI-TOF mass spectra in the present study are star polymers with lophine end groups and thus we presented MALDI-TOF spectrum of star-PBA_{Lophine}. We did not synthesize star-PBA_{HABI} but synthesized network-PBA_{HABI}. Measuring MALDI-TOF mass spectra of such network polymers is, in general, impossible. Since our final products are network polymers, such spectral analyses for them are difficult.

Other Suggestions

It is found that azo-benzene contained polymers, which exhibit photo induced T_g transformation between liquid and solid state induced by cis-trans exchange of azo-benzene has been reported by Wu in a recent paper (Title: Photoswitching of glass transition temperatures of azobenzene-containing polymers induces reversible solid-to-liquid transitions, Nature Chemistry (2016) doi:10.1038/nchem.2625). These results covers the new developments of azo-benzene based materials. We suggest this paper should be cited.

Response

The reference has been cited in the revised manuscript.

For Reviewer 3:

Comment

The authors have analyzed solution or dispersion state of the network polymers by means of spectroscopic measurement, GPC measurement and NMR measurement. Because the network polymer is usually insoluble, it indicates containing of a soluble component composed of incomplete-network polymer. In fact the authors discuss the intramolecular dimerization in the manuscript. If the soluble component is a main part of the polymer, the influence of thermal plasticization on liquidity should be also taken into account. Relatively strong light source (like 150mwcm⁻²) used here is often increase the temperature.

Since control of the liquid-solid state of the substance is a basic part related to materials science, a lot of people will have interest in this subject. The concept is different from previous cases. Therefore, if the authors make it clear in their response that the influence of the temperature rise by light irradiation is negligible, I recommend the publication. In this case, it is necessary to reconsider the description at line 29-31, because there are recently reported on polymer systems (Nature Chemistry (2016) doi:10.1038/nchem.2625, Journal of Adhesion (2016)

doi.org/10.1080/00218464.2016.1219255)

Response

We are grateful to the comment and have carefully examined the dependence of temperature on the change in physical state. Thus, temperature-dependent dynamic mechanical analysis (DMA) was performed using Anton Paar MCR 702 equipped with a UV irradiation option and a temperature controller, where the measurement temperature was set to 25 °C for **network-PDMS_{HABI}** and to 50 °C for **network-PBA_{HABI}**. Based on DMA, UV-triggered liquefaction and loss of liquidity by terminating UV was clearly confirmed and change of the temperature under the irradiation of UV was negligible under the irradiation condition. The related discussion has been added to the revised manuscript. Also, the references have been cited in the revised manuscript.

Reviewers' comments:

Reviewer #1 (Remarks to the Author):

Authors have made a serious effort to revise and improve the manuscript. All my comments have been addressed adequately. However, one question has been appeared to me. can you judge the sample is in "nonliquid state" even G' is always greater than G'' ? If it is in mayonnaise-like state, G' should be greater than G'' at smaller strain region? It would be helpful for readers if authors give a general definition (or examples) of nonliquid mayonnaise-like state by using these G' and G'' values. If this point becomes clear, the manuscript can be published.

Reviewer #2 (Remarks to the Author):

Q1. According to the GPC results provided by the authors, the molecular weight of network-PDMSHABI is estimated to be 6100 (Mn); 11200 (Mp) and star-PDMSLophine is 3700 (Mn); 4800 (Mp), respectively. Based on these results, the molecular weight of network-PDMSHABI is only ~2 times (using value of Mn is 1.65 and Mp is 2.33) compared with star-PDMSLophine. These features are rather different from the "polymer network" proposed by the authors in their paper (for direct image see Figure 1 and Figure 3). Additionally, the GPC values of network-PBAHABI should be provided in supporting information in page 7.

Q2. From the results of GPC analysis, the molecular weight of network-PDMSHABI is estimated to be ~6000 (Mn) or ~11200 (Mp). We believe this mass distribution could be done by MALDI-TOF analysis, or at least tried.

Q3. It should be pointed out that the oxidation of the lophine to HABI is not complete in this work based on the ¹H NMR results presented by the authors (network-PBAHABI and network-PDMSHABI are calculated to be 60% and 70%, respectively). Without oxidation, lophine can't connect with each other by chemical bond and remain as "dead end" in the polymers. Thus the conversion efficiency from lophine to HABI is vital to the argument of this paper. However, in this work, the oxidation is not complete and there is high fraction of lophine units remaining in the polymer (more than 30%). These might indicate that the claimed network of these polymers can hardly be accurate as proposed by the authors, which should be considered carefully.

Q4. In Figure 3, the authors are trying to demonstrate the concentrate effect on the reconstruction from the star-PBATPIR to network-PBAHABI. In our former review report, we try to express that there are two factors involved in this experiment, i.e. UV light and concentration. It seems as if the GPC would not change in various concentrations without UV irradiation. Strictly speaking, this should be done as a blank reference to make their argument indisputable. That is the GPC spectra of network-PBAHABI in 1, 5, 10, 50 and 100 mg/ml without UV irradiation should be added.

Q5. Based on the ¹H NMR results (see Figure S8), we come to find that solvent residual peaks of DMF and hexane are clearly observed. It should be noted that the solvent remaining could have an effect, as the authors are trying to argue their materials as "Photo-triggered solvent-free metamorphosis of polymeric materials" (title). We suggest this issue should be considered and discussed.

Q6. In Figure S8, the -NH hydrogen of triphenylimidazole was still defined as "f", whose chemical shift is around ~9 from the figure. As we have pointed out in our former review report, the -NH of triphenylimidazole is generally around 12~13. We can also see this characters from the results in Figure S4 and Figure S9 from the authors. In ¹H NMR spectrum, that is a huge difference! Since the -NH from imidazole reflect the oxidation in the synthesis of HABI directly, we believe this information is very important for HABI based materials. The authors should explain this point

clearly. Once again, we suggest the integral area of the peaks in ^1H NMR spectrum presented by the authors should be marked and added, which would help the further discussion on this issue.

Q7. In Figure 2(d), the amount of TPIRs are defined as 0% based on the absorbance at 601 nm by the authors. Based on previous investigations, the absorbance at 601 nm should be attributed to TPIRs (triphenylimidazole radicals), which means before UV irradiation, there is no absorbance at 601 nm, ie no EPR signals. However, these results would contradict with Figure 2(c). We suppose the defining by the authors is for convenience while our concern is that this issue might be misleading for common readers.

Q8. The authors have presented a polymer network using the reversible photochromism of hexaarylbiimidazole (HABI) as photo-responsive metamorphosis polymeric materials. This area has been widely investigated using azo-benzene based materials. These materials, either small molecule or more recently, polymer materials, have been demonstrated for the same purpose. Generally, they could be easily synthesized with great reversibility. Fairly speaking, the phenomenon is far from new. I am quite curious on the advantage of this system, especially compared with azo-benzene materials, for example, like better performance, easier to synthesize, greater reversibility, or something else.

Other mistakes:

Page 9. In the synthesis of "Synthesis of network-PDMSHABI" "After drying under reduced pressure, an elastic solid poorly soluble to organic solvents (network-PBAHABI) was obtained....." should be network-PDMSHABI

Reviewer #3 (Remarks to the Author):

From the measurement results of dynamic viscoelasticity, the authors clarified that fluidity increases with light irradiation at a constant temperature. So my doubt ceased. I think this concept is very interesting. Regarding the additional data, what's worrisome is that the initial values of G' and G'' of network polymers are very low. Especially it seems strange that G'' is larger than G' . I think it is necessary more discussions based on previous research examples, expert opinion, etc.

Point-to-Point Answers to the Referees' Comments.

We are grateful to all of the reviewers for giving us valuable comments again. We have strongly encouraged and further elaborated our manuscript. The text revised according to the present comments are highlighted in light green.

For Reviewer #1:

Comment

Authors have made a serious effort to revise and improve the manuscript. All my comments have been addressed adequately. However, one question has been appeared to me. Can you judge the sample is in "nonliquid state" even G'' is always greater than G' ? If it is in mayonnaise-like state, G' should be greater than G'' at smaller strain region? It would be helpful for readers if authors give a general definition (or examples) of nonliquid mayonnaise-like state by using these G' and G'' values. If this point becomes clear, the manuscript can be published.

Response

We are very grateful to the comment.

From the viewpoint of rheology, our materials show higher G'' than G' and this characteristic is typical for liquid materials whereas our materials were not flowable without UV irradiation. In fact, to define solid or liquid with such materials is very difficult. Even for an inorganic glass, some researchers in physics consider the glass is in liquid state and frequently argued with the researchers who think that the glass is in solid state. Relating researches have continuously been reported (e.g., Biroli et al., Breakdown of elasticity in amorphous solids, *Nat. Phys.* **12**, 1130–1133 (2016), DOI: 10.1038/NPHYS3845, Cooper et al., Irreversible reorganization in a supercooled liquid originates from localized soft modes, *Nat. Phys.* **4**, 711–715 (2008), DOI:10.1038/nphys1025, and Dunleavy et al., Mutual information reveals multiple structural relaxation mechanisms in a model glass former, *Nat. Commun.* **6**, 6089 (2014), DOI: 10.1038/ncomms7089) but to elucidate it remains one of the biggest challenges in physics.

On the other hand, for polymer materials, so called “the pitch drop experiment (Edgeworth et al., *Eur. J. Phys* 198–200 (1984))” started from 1927 and still continues is suggestive. We generally think that a plastic material is solid but the above study tells us that the plastic material is liquid with the time scale of about 10 years (fall of one drop of the plastic occurs per 10 years). Therefore, it depends on time scale, i.e., frequency-dependent DMA measurements will be important whether the material can be regarded as solid or liquid from the experimental chemistry aspect. Although we promptly requested frequency-dependent DMA measurements for our materials to Anton Paar Inc. again, the measurements have been found difficult because of machine time problem. Instead, they kindly provided data of frequency dependence of polysiloxane-based materials as examples, which are commercially available famous toys called *Strong Gum* and *Blobimals* (A youtube video has also

been introduced; <https://www.youtube.com/watch?v=g0uhVt70MLM>). These materials are nonliquid for hours, but gradual deformation occurs with standing overnight. From frequency-dependent DMA measurements for these materials, G' 's are higher than G'' 's with the frequency over 5 Hz or 9 Hz, but G'' 's become higher than G' 's with the frequency below 5 Hz (Figure P1).

Figure P1 | Frequency-dependent plots of G' and G'' for the polysiloxane-based materials (blue: Blobimals, and red: Strong Gum)

Moreover, rheological properties of bottlebrush polymer melts and networks under solvent-free condition have recently been reported (Daniel et al., Solvent-free, supersoft and superelastic bottlebrush melts and networks, *Nat. Mater.* **15**, 183–190 (2016), DOI: 10.1038/NMAT4508). While the above silicon-based *Blobimals* show only one intersection in the curves of G' and G'' , interestingly, such bottlebrush polymer melts and networks show several intersections in their frequency-dependent curves of G' and G'' . We envision that our materials will show such curious frequency dependence with G' and G'' . Although frequency-dependent DMA measurements cannot be attained this time because of the machine time problem and are beyond the scope of the present study, we will report such studies in the future, inspired and encouraged by the above valuable comment.

This point is indeed an essence of science for our study and the related discussions have been described in the revised manuscript.

For Reviewer #2:

Comment #1

According to the GPC results provided by the authors, the molecular weight of

network-PDMSHABI is estimated to be 6100 (Mn); 11200 (Mp) and star-PDMSLophine is 3700 (Mn); 4800 (Mp), respectively. Based on these results, the molecular weight of network-PDMSHABI is only ~2 times (using value of Mn is 1.65 and Mp is 2.33) compared with star-PDMSLophine. These features are rather different from the “polymer network” proposed by the authors in their paper (for direct image see Figure 1 and Figure 3). Additionally, the GPC values of network-PBAHABI should be provided in supporting information in page 7.

Response

Although complete network polymers are, in general, insoluble in solvents, solvent-soluble part of network polymers containing precursor polymers and intermolecularly reacted products with moderate molecular weights can be detectable with GPC as reported elsewhere. As such, we described “The GPC chromatogram of the THF soluble part of **network-PDMS_{HABI}**” in the results and discussion. Since the GPC chromatogram reflects only the THF soluble part of **network-PDMS_{HABI}**, we believe the results are reasonable.

The GPC values of “THF soluble part” of as-prepared **network-PDMS_{HABI}** has been added in the revised manuscript in page 7.

Comment #2

From the results of GPC analysis, the molecular weight of network-PDMSHABI is estimated to be ~6000 (Mn) or ~11200 (Mp). We believe this mass distribution could be done by MALDI-TOF analysis, or at least tried.

Response

Similar to the response to Comment #1, GPC analysis for **network-PDMS_{HABI}** tells us only the molecular weight and molecular weight distribution of “solvent soluble part” and MALDI-TOF mass analysis of the solvent-soluble part does not reflect whole of the products. We have tried MALDI-TOF mass analysis for the solvent-soluble part of **network-PDMS_{HABI}**, yet the analysis was found difficult.

Comment #3

It should be pointed out that the oxidation of the lophine to HABI is not complete in this work based on the 1H NMR results presented by the authors (network-PBAHABI and network-PDMSHABI are calculated to be 60% and 70%, respectively). Without oxidation, lophine can't connect with each other by chemical bond and remain as “dead end” in the polymers. Thus the conversion efficiency from lophine to HABI is vital to the argument of this paper. However, in this work, the oxidation is not complete and there is high fraction of lophine units remaining in the polymer (more than 30%). These might indicate that the claimed network of these polymers can hardly be accurate as proposed by the authors, which should be considered carefully.

Response

Owing to the valuable comments for the previous version of our manuscript (Comment #7 for the previous version), we could evaluate the conversion yield from TPIRs to HABIs. Although lophine units are remained ca. 40% for **network-PBA_{HABI}** and ca. 30% for **network-PDMS_{HABI}** as described in the manuscript, the existence of lophine units hardly affected the conclusion of our repeatable MAT strategy as revealed by the combination of GPC and DMA analyses. In addition, we have carefully considered the existence of unreacted end groups from the beginning of this study as illustrated in the middle of Figure 3b, because it is well-known that unreacted end groups and intramolecularly reacted loops inherently remain in network polymers formed by reacting end groups of star polymers and thus such network polymers can be called as elastomers.

Comment #4

In Figure 3, the authors are trying to demonstrate the concentrate effect on the reconstruction from the star-PBATPIR to network-PBAHABI. In our former review report, we try to express that there are two factors involved in this experiment, i.e. UV light and concentration. It seems as if the GPC would not change in various concentrations without UV irradiation. Strictly speaking, this should be done as a blank reference to make their argument indisputable. That is the GPC spectra of network-PBAHABI in 1, 5, 10, 50 and 100 mg/ml without UV irradiation should be added.

Response

GPC measurements of **network-PBA_{HABI}** in 1, 5, 10, 50 and 100 mg/mL without UV irradiation are difficult because network polymers are generally insoluble in a solvent. As such, we described “The mechanism of the reaction was investigated using three concentrations (1, 5, and 10 mg/mL) of **network-PBA_{HABI}** dispersions in THF. UV irradiation to the dispersions completely dissolved **network-PBA_{HABI}** in THF and the solution became blue.” in the beginning of the second paragraph in page 6. It is not until UV was irradiated to the mixtures that we could obtain the solutions of them.

Comment #5

Based on the ¹H NMR results (see Figure S8), we come to find that solvent residual peaks of DMF and hexane are clearly observed. It should be noted that the solvent remaining could have an effect, as the authors are trying to argue their materials as “Photo-triggered solvent-free metamorphosis of polymeric materials” (title). We suggest this issue should be considered and discussed.

Response

By calculating the integration of the solvent derived signals in ¹H NMR, we confirmed that the amount of the solvents in the polymer sample is small and the effect of them is negligible for the present IRLNC experiments as clearly supported by the data.

Comment #6

In Figure S8, the –NH hydrogen of triphenylimidazole was still defined as “f”, whose chemical shift is around ~9 from the figure. As we have pointed out in our former review report, the –NH of triphenylimidazole is generally around 12~13. We can also see this characters from the results in Figure S4 and Figure S9 from the authors. In ¹H NMR spectrum, that is a huge difference! Since the –NH from imidazole reflect the oxidation in the synthesis of HABIs directly, we believe this information is very important for HABI based materials. The authors should explain this point clearly. Once again, we suggest the integral area of the peaks in ¹H NMR spectrum presented by the authors should be marked and added, which would help the further discussion on this issue.

Response

As was mentioned in the response for Comment #7 for previous version, generally, PDMS is insoluble in DMSO, MeOH, and other polar solvents but soluble, for example, in CHCl₃ and hexane and partially in acetone. Given that there exist some acetone insoluble parts, some of the products possibly form a kind of small molecular aggregates like micelles. Therefore, no wonder there exist a huge difference between Figure S4 and Figure S9 because such molecular aggregates cause shift of signals in ¹H NMR spectra.

In addition, the imidazole-derived –NH signal generally appears at around 12–13 ppm in case DMSO-*d*₆ is used as the solvent, however, disappearance or shift of the signal by using less polar deuterated solvent is also frequently observed as reported elsewhere. For example, –NH signals of lophine-tethered molecules measured using CDCl₃ appeared at around 9 ppm in their ¹H NMR spectra (*Macromolecules* **43**, 3764–3769 (2010)), which is comparable to our study.

Comment #7

In Figure 2(d), the amount of TPIRs are defined as 0% based on the absorbance at 601 nm by the authors. Based on previous investigations, the absorbance at 601 nm should be attributed to TPIRs (triphenylimidazole radicals), which means before UV irradiation, there is no absorbance at 601 nm, ie no EPR signals. However, these results would contradict with Figure 2(c). We suppose the defining by the authors is for convenience while our concern is that this issue might be misleading for common readers.

Response

As described in the figure caption, Figure 2(c) shows that “ESR spectra of **network-PBA_{HABI}**” and thus the **bulk materials** were subjected to the ESR measurements. On the other hand, Figure 2(d) shows that “Time-dependent plots of the amounts of TPIR in a THF solution of **network-PBA_{HABI}** (10 mg/mL)” as also described in the figure caption and thus **THF solutions** were subjected to UV measurements. We do not think that the comparison of above two data is not so beneficial because

measurements for bulk material and its solution states are completely different matters and we believe that the definition of the amount of TPIRs is accordingly appropriate.

Moreover, as discussed in page 6, the recoupling reaction between TPIRs should be retarded along with the progress of network formation because it is generally difficult for polymer chains to diffuse in the network matrix particularly in the later stage of network formation reaction. Since, UV irradiation to the dispersion of **network-PBA_{HABI}** and THF resulted in its THF solutions, the remained TPIR end groups probably reacted each other in solution state and thus it is reasonable that the UV spectrum shows negligible absorbance at 601 nm.

Comment #8

The authors have presented a polymer network using the reversible photochromism of hexaarylbimidazole (HABI) as photo-responsive metamorphosis polymeric materials. This area has been widely investigated using azo-benzene based materials. These materials, either small molecule or more recently, polymer materials, have been demonstrated for the same purpose. Generally, they could be easily synthesized with great reversibility. Fairly speaking, the phenomenon is far from new. I am quite curious on the advantage of this system, especially compared with azo-benzene materials, for example, like better performance, easier to synthesize, greater reversibility, or something else.

Response

The present reversible MAT strategy is completely different conventional studies. As was mentioned in the introduction, tuning of elasticity by crystallization–crystal melting of azobenzene-containing small molecules is difficult. On the other hand, tuning of glass transition temperature (T_g) by photo-isomerization of azobenzene-containing polymers (azopolymers) overcomes the above problem (*Nat. Chem.* **9**, 145-151 (2017)). Although the easiest methodology for functionalizing polymer materials is to change side chains of polymers, the side chains of the above azopolymers have already been occupied by azobenzene-containing chains themselves. In contrast, our methodology is, in principle, applicable irrespective of the types of polymers and side chain functionalization will be attained by applying easily available side chain reactive liquid polymers such as poly(methylhydrosiloxane), poly(dimethylsiloxane-co-(3-aminopropyl)methylsiloxane), and poly(dimethylsiloxane-co-(2-(3,4-epoxycyclohexyl)ethyl)methylsiloxane). In fact, all of their linear polymers are commercially available and we think that present synthetic means for PDMS can be extendable to their star-shaped polymers. Therefore, we believe that our repeatable MAT strategy has significant advantage against conventional ones.

Comment #9

Other mistakes: Page 9. In the synthesis of “Synthesis of network-PDMSHABI” “After drying under reduced pressure, an elastic solid poorly soluble to organic solvents (network-PBAHABI) was

obtained.....” should be network-PDMSHABI

Response

We are grateful to this comment. The typo in the supplementary information (page 9) has been revised.

For Reviewer #3:

Comment

From the measurement results of dynamic viscoelasticity, the authors clarified that fluidity increases with light irradiation at a constant temperature. So my doubt ceased. I think this concept is very interesting. Regarding the additional data, what's worrisome is that the initial values of G' and G'' of network polymers are very low. Especially it seems strange that G'' is larger than G' . I think it is necessary more discussions based on previous research examples, expert opinion, etc.

Response

We are grateful to the comment and are strongly encouraged. The Reviewer #1 raised the same comment concerning to an interpretation of DMA results. The rheological aspects of our materials is very interesting from the viewpoint in physics and the above comment “Especially it seems strange that G'' is larger than G' .” is essence of science with our materials. The related detailed discussion has been described in the revised version of our manuscript.

For further information, the response to Reviewer #1 is appended below.

From the viewpoint of rheology, our materials show higher G'' than G' and this characteristic is typical for liquid materials whereas our materials were not flowable without UV irradiation. In fact, to define solid or liquid with such materials is very difficult. Even for an inorganic glass, some researchers in physics consider the glass is in liquid state and frequently argued with the researchers who think that the glass is in solid state. Relating researches have continuously been reported (e.g., Biroli et al., Breakdown of elasticity in amorphous solids, *Nat. Phys.* **12**, 1130-1133 (2016), DOI: 10.1038/NPHYS3845, Cooper et al., Irreversible reorganization in a supercooled liquid originates from localized soft modes, *Nat. Phys.* **4**, 711-715 (2008), DOI:10.1038/nphys1025, and Dunleavy et al., Mutual information reveals multiple structural relaxation mechanisms in a model glass former, *Nat. Commun.* **6**, 6089 (2014), DOI: 10.1038/ncomms7089) but to elucidate it remains one of the biggest challenges in physics.

On the other hand, for polymer materials, so called “the pitch drop experiment (Edgeworth et al., *Eur. J. Phys* 198-200 (1984))” started from 1927 and still continued is suggestive. We generally think that a plastic material is solid but the above study tells us that the plastic material is liquid with the time scale of about 10 years. Therefore, it depends on time scale, i.e., frequency-dependent DMA

measurements will be important whether the material can be regarded as solid or liquid from the experimental chemistry aspect. Although we promptly requested frequency-dependent DMA measurements for our materials to Anton Paar Inc. again, the measurements have been found difficult because of machine time problem. Instead, they kindly provided data of frequency dependence of polysiloxane-based materials as examples, which are commercially available famous toys called *Strong Gum* and *Blobimals* (A youtube video has also been introduced; <https://www.youtube.com/watch?v=g0uhVt70MLM>). These materials are nonliquid for hours, but gradual deformation occurs with standing overnight. From frequency-dependent DMA measurements for these materials, G' 's are higher than G'' 's with the frequency over 5 Hz or 9 Hz, but G'' 's become higher than G' 's with the frequency below 3 Hz (Figure P1).

Figure P1 | Frequency-dependent plots of G' and G'' for the polysiloxane-based materials (blue: Blobimals, and red: Strong Gum)

Moreover, rheological properties of bottlebrush polymer melts and networks under solvent-free condition have recently been reported (Daniel et al., Solvent-free, supersoft and superelastic bottlebrush melts and networks, *Nat. Mater.* **15**, 183-190 (2016), DOI: 10.1038/NMAT4508). While the above silicon-based *Blobimals* show only one intersection in the curves of G' and G'' , interestingly, such bottlebrush polymer melts and networks show several intersections in their frequency-dependent curves of G' and G'' . We envision that our materials will show such curious frequency dependence with G' and G'' . Although frequency-dependent DMA measurements cannot be attained this time because of the machine time problem and are beyond the scope of the present study, we will report such studies in the future, inspired and encouraged by the above valuable comment.

This point is indeed an essence of science for our study and the related discussions have been

described in the revised manuscript.

Reviewers' comments:

Reviewer #1 (Remarks to the Author):

Authors have made a serious effort to revise and improve the manuscript. All my comments have been addressed adequately. I recommend publication as is.

Reviewer #2 (Remarks to the Author):

The issues raised by the referees have been addressed.
I suggest the paper can be accepted.

Reviewer #3 (Remarks to the Author):

If frequency dependence is the reason, the authors should show data on viscoelastic properties in the high frequency range, which is direct evidence. Readers who have read the added explanation will be wondering why this data is not displayed. On the other hand, since it is communication, it may be omitted if there is a valid reason that, for example, precise values can not be measured immediately or easily for any reason, and so on. However, since it takes less time to measure the frequency dependence, the limitation of the machine time currently in the comment is inappropriate as a reason.

Point-to-Point Answers to the Referees' Comments.

We are grateful to the comment and indeed we could increase the significance of our manuscript by the comment. The text revised according to the comment was highlighted in light blue.

For Reviewer #3:**Comment**

If frequency dependence is the reason, the authors should show data on viscoelastic properties in the high frequency range, which is direct evidence. Readers who have read the added explanation will be wondering why this data is not displayed. On the other hand, since it is communication, it may be omitted if there is a valid reason that, for example, precise values cannot be measured immediately or easily for any reason, and so on. However, since it takes less time to measure the frequency dependence, the limitation of the machine time currently in the comment is inappropriate as a reason.

Response

We have obtained a special opportunity for retrying DMA analysis with kind cooperation with Anton Paar Inc. and have conducted frequency-dependent measurements of G' , G'' , and complex viscosity (η^*). We have reached the acceptable conclusion for our study and the related discussion has been described in the revised manuscript.

REVIEWERS' COMMENTS:

Reviewer #3 (Remarks to the Author):

I had expected that the viscosity change due to light irradiation would be larger in the higher frequency range, but it was not. The current result seems not to be direct evidence. As pointed out by the authors, it is conceivable that the reaction did not progress uniformly throughout the whole due to, for example, the thickness of the sample. It can be said that there are parts that are not at least clear. I recommend that the manuscript is published in nature communication by considering that they studied as far as they can at this time. I hope the mechanism will become clearer in subsequent papers.

Point-to-Point Answers to the Referees' Comments.

For Reviewer #3:

Comment

I had expected that the viscosity change due to light irradiation would be larger in the higher frequency range, but it was not. The current result seems not to be direct evidence. As pointed out by the authors, it is conceivable that the reaction did not progress uniformly throughout the whole due to, for example, the thickness of the sample. It can be said that there are parts that are not at least clear. I recommend that the manuscript is published in nature communication by considering that they studied as far as they can at this time. I hope the mechanism will become clearer in subsequent papers.

Response

We consider that the similar change of viscosity at lower and higher frequency ranges could not give any serious influence on the validity of the present IRLNC system and we have devoted utmost effort for analyzing and characterizing properties of the polymers with the present system. Expanding the scope of available polymers and of cleavable and reformable covalent bonds and a systematic study for characterizing their properties have a possibility for elucidating a more detailed IRLNC mechanism and such studies will be reported in the near future.